# Antiviral Activities of Mastoparan-L-Derived Peptides against Human Alphaherpesvirus 1

**DOI:** 10.3390/v16060948

**Published:** 2024-06-12

**Authors:** Liana Costa Pereira Vilas Boas, Danieli Fernanda Buccini, Rhayfa Lorrayne Araújo Berlanda, Bruno de Paula Oliveira Santos, Mariana Rocha Maximiano, Luciano Morais Lião, Sónia Gonçalves, Nuno C. Santos, Octávio Luiz Franco

**Affiliations:** 1Pós-Graduação em Patologia Molecular, Campus Darcy Ribeiro, Universidade de Brasília, Brasília 70910-900, DF, Brazil; 2Centro de Análises Bioquímicas e Proteômicas, Pós-Graduação em Ciências Genômicas e Biotecnologia, Universidade Católica de Brasília, Brasília 70790-760, DF, Brazil; 3Pós-Graduação em Biotecnologia, Universidade Católica Dom Bosco, Campo Grande 79117-900, MS, Brazil; 4Laboratório de Ressonância Magnética Nuclear, Instituto de Química, Universidade Federal de Goiás, Goiânia 74690-900, GO, Brazil; 5Instituto de Medicina Molecular, Faculdade de Medicina, Universidade de Lisboa, 1649-028 Lisbon, Portugalnsantos@fm.ul.pt (N.C.S.)

**Keywords:** antiviral peptide, mastoparan, peptide–membrane interaction

## Abstract

Human alphaherpesvirus 1 (HSV-1) is a significantly widespread viral pathogen causing recurrent infections that are currently incurable despite available treatment protocols. Studies have highlighted the potential of antimicrobial peptides sourced from *Vespula lewisii* venom, particularly those belonging to the mastoparan family, as effective against HSV-1. This study aimed to demonstrate the antiviral properties of mastoparans, including mastoparan-L [I^5^, R^8^], mastoparan-MO, and [I^5^, R^8^] mastoparan, against HSV-1. Initially, Vero cell viability was assessed in the presence of these peptides, followed by the determination of antiviral activity, mechanism of action, and dose-response curves through plaque assays. Structural analyses via circular dichroism and nuclear magnetic resonance were conducted, along with evaluating membrane fluidity changes induced by [I^5^, R^8^] mastoparan using fluorescence-labeled lipid vesicles. Cytotoxic assays revealed high cell viability (>80%) at concentrations of 200 µg/mL for mastoparan-L and mastoparan-MO and 50 µg/mL for [I^5^, R^8^] mastoparan. Mastoparan-MO and [I^5^, R^8^] mastoparan exhibited over 80% HSV-1 inhibition, with up to 99% viral replication inhibition, particularly in the early infection stages. Structural analysis indicated an α-helical structure for [I^5^, R^8^] mastoparan, suggesting effective viral particle disruption before cell attachment. Mastoparans present promising prospects for HSV-1 infection control, although further investigation into their mechanisms is warranted.

## 1. Introduction

Human alphaherpesvirus 1, also called herpes simplex virus type 1 (HSV-1), is a Herpesviridae family member, along with HSV-2 and varicella-zoster virus (VZV) that causes varicella (chicken pox) and herpes zoster [1]. HSV-1 is known to cause cold sores and painful orofacial lesions in primary infections, persisting in the host by subsequent sensory neurons and ganglia infection [2]. In the worst cases, the disease can spread to the central nervous system, leading to meningitis and encephalitis, especially in newborns [3]. Recent data indicate that 3.7 billion people below 50 years old carry HSV-1 [4,5], and approximately 500 million people between the ages of 17 and 49 are infected with HSV-2 [6,7]. These viruses have been studied for some years, and there is a well-established treatment protocol using nucleoside analogs, such as acyclovir and derived prodrugs. However, the emergence of viral resistance is a worrying factor [8], especially for immunocompromised patients [9,10]. For this reason, new molecules derived from and inspired by naturally occurring products are promising tools for infection control.

Several studies have demonstrated the potential of antiviral peptides (AVPs) derived from different sources, from invertebrates to vertebrates [11,12,13]. AVPs share standard features with other antimicrobial peptides, such as non-specific activity, small size (typically 10 to 50 amino acid residues), positive charge, amphipathicity, and diverse structures, including α-helix, β-sheet, extended helix, and loops [14]. AVPs are primarily known as virucidal agents, exerting their action through direct viral particle inactivation or competition for binding sites on the host cell membrane proteins. This disruption of fundamental interactions ultimately leads to viral adsorption and cell penetration entry prevention to the target cells. Additionally, AVPs may exert antiviral effects at other viral cycle stages, such as suppressing viral gene expression, thus offering potential therapeutic benefits across various steps of viral infection [15].

Arthropods’ venoms are a rich source of AVPs, including melittin from bees [16], Hp1239 and Hp1036 from scorpions [17], alloferon 1 and 2, along with their analogs from blowflies [18], and lactarcin one from spiders [19]. Among them, peptides from the mastoparan (MP) family, derived from wasp venom, have already shown antibacterial, antifungal, and antiviral activity [11,20]. MP adopts an amphipathic α-helical structure, has 14 amino acid residues, and is known to mediate mast cell degranulation and histamine release [21]. In previous works using bioinformatics tools, modifications were introduced in the mastoparan L (mast-L) peptide, resulting in [I^5^, R^8^] mastoparan [22] and mastoparan-MO (mast-MO) [23]. Minor adjustments were made in the mast-L primary structure to decrease its hemolytic activity and cytotoxicity while enhancing antimicrobial activity. Thus, two alanine residues were replaced by an isoleucine in the fifth position and an arginine in the eighth position, hence the name [I^5^, R^8^] mastoparan [22]. The study showed broad-spectrum antimicrobial activity against bacteria and fungi, low cytotoxicity, and no hemolytic activity toward human erythrocytes. Furthermore, experiments using fluorescent probes in two bacterial strains have shown that this peptide can permeabilize membranes. This conclusion was further supported by scanning electron microscopy images, which clearly show significant bacterial membrane damage when exposed to [I^5^, R^8^] mastoparan [22].

Mast-MO was developed by studying immunomodulatory peptide sequences. To enhance its immune-modulating properties, a five-amino-acid residue motif (FLPII) found in other peptides was added to the N-terminal mast-L sequence [23]. The results showed that mast-MO had lower cytotoxicity and higher antimicrobial activity when compared with its parental peptide, mast-L. Moreover, mast-MO showed an anti-inflammatory profile in in vivo assays with mice by reducing the release of pro-inflammatory cytokines and, at the same time, improving leukocyte migration to the infection site [23].

Here, we show that the previously designed peptides tested against HSV-1 exerted a virucidal effect. We present additional information about the structure of a mastoparan-derived peptide, [I^5^, R^8^] mastoparan.

## 2. Materials and Methods

### 2.1. Cell Culture and Viruses

Vero cells (*Cercopithecus aethiops* monkey kidney cells) were obtained from the Núcleo de Virologia of the Central Laboratory in Brazil’s Federal District (LACEN/DF). The cells were cultivated in Dulbecco’s Eagle minimum essential medium (DMEM) was supplemented with L-glutamine, streptomycin/penicillin antibiotics, and 10% heat-inactivated fetal bovine serum (FBS), and incubated at 37 °C in a humidified atmosphere containing 5% CO_2_ [12]. The HSV-1 strain (clinical isolate) was provided by the Federal University of Rio de Janeiro collection. The acyclovir (ACV) powder was purchased from Sigma-Aldrich. The ACV stock solution was prepared for experimentation using ultrapure water at 3.2 mg.mL^−1^.

### 2.2. Synthetic Peptides and Lipids

Mastoparan-MO (FLPIIINLKALAALAKKIL-NH_2_), mastoparan L (INLKALAALAKKIL-NH_2_), and [I^5^, R^8^] mastoparan (INLKILARLAKKIL-NH_2_) were synthesized and purchased from AminoTech (Sorocaba, SP, Brazil). All peptides were used after verifying the purity and molecular masses by mass spectrometry (MALDI-ToF/ToF Ultraflex III, Bruker Daltonics (Billerica, MA, USA)). The purity for all peptides was higher than 95%, and the quantification was carried out according to Murphy and Kies [24]. POPC (1-palmitoyl-2-oleyl-*sn*-glycero-3-phosphocholine) and POPS (1-palmitoyl-2-oleyl-*sn*-glycero-3-phospho-L-serine) were provided by Avanti Polar Lipids (Alabaster, AL, USA). Sigma-Aldrich (St. Louis, MO, USA) provided cholesterol (Chol), DPH (1,6-diphenyl-1,3,5-hexatriene), and TMA-DPH (1-(4-trimethylammoniumphenyl)-6-phenyl-1,3,5-hexatriene p-toluenesulfonate). The buffer utilized in the studies as working solution was ten mM HEPES pH 7.4, with 150 mM NaCl.

### 2.3. Cytotoxicity Assays

Vero cell monolayers were grown in 96-well microplates to determine the maximum non-toxic peptide concentration (MNTC). The cells were exposed to two-fold serial solutions of peptide (200, 100, 50, 25, 12.5, 6.25, and 3.12 µg.mL^−1^) in triplicate and incubated for 48 h at 37 °C. After incubation, the cells were examined using an inverted optical microscope, evaluate with the aim of evaluating the morphological alterations. Cell viability was further assessed by 3-(4,5-dimethyl thiazolyl)-2,5-diphenyl-tetrazoliumbromide (MTT; Sigma) at 5 µg.mL^−1^. Absorbance measurements at 490 nm were made on a BioTek PowerWave HT microplate spectrophotometer [12].

### 2.4. Viral Titration

To determine the 50% tissue culture infective dose (TCID_50_) of the HSV-1 sample, tenfold serial dilutions were prepared and added to Vero cell monolayers previously incubated in 96-well microplates. Each serial dilution was inoculated six times in the microplate and incubated at 37 °C in 5% CO_2_ for 48 h. After this period, each well was analyzed for viral cytopathic effect. Subsequently, the calculation proposed by Reed and Muench [25] was performed.

### 2.5. Antiviral Activity by Plaque Assay

Antiviral activity was quantified by plaque assay as follows. Vero cell monolayers previously incubated in 24-well microplates were treated with the MNTC of each peptide and inoculated with 100 µL of HSV-1 suspension at 100 TCID_50_ (10^−5.45^) and after that, incubated at 37 °C in 5% CO_2_ for 48 h. After a freeze/thaw cycle, tenfold serial dilutions of each well were prepared. Then, 500 µL of each dilution were added (in triplicate) on Vero cell monolayers (previously incubated in 6-well microplates, incubated at 37 °C in 5% CO_2_, for 1 h). After the incubation period, the inoculum was removed, and a carboxymethylcellulose overlay medium (DMEM, 2% FBS, 1% CMC) was added to each well, followed by another incubation for 48 h. Subsequently, 3 mL of 10% formaldehyde fixation solution was added to each well and left at room temperature for at least 90 min. Then, the CMC/fixation solution was removed, and the microplate was washed with water. Finally, to count the plaque formation units (PFUs), 600 µL of crystal violet 1% stain was added to each well for 15 min, then removed and washed with water [26]. To perform plaque counting of each well, the average plaque numbers were registered and applied to the following Equation (1):(1)$$PFU/mL=\frac{Avg.\;of\;plaques}{D\times V}$$
where D is the dilution, and V is the volume of the diluted virus added to the plate [27].

### 2.6. Inhibition of Replication Steps

Microtubes were prepared with the peptides at the MNTC and 100 µL of HSV-1 suspension at 100 TCID_50_ to infer if the peptide could inactivate the viral particle before infection (Appendix A). Immediately afterward, the microtubes (peptide + virus and positive control with just the viral suspension) were incubated for 90 min at 37 °C, and antiviral activity was determined by plaque assay. To assess whether the peptides could bind with the cellular receptors used by the virus (Appendix A), Vero cell monolayers previously incubated in 24-well microplates were first treated with peptide solutions at the MNTC for 1 h at 4 °C. Immediately after that, they were washed with DMEM without serum to remove the free peptide, and 100 µL of HSV-1 suspension at 100 TCID_50_ was added to both treated and untreated cultures and again incubated at 37 °C for 48 h. Later, the microplate was frozen for cell lysis, followed by plaque assay (item 2.5). Moreover, to evaluate if the peptide was able to interfere with viral penetration entry, Vero cell monolayers previously incubated in 24-well microplates were first inoculated with 100 µL of HSV-1 suspension at 100 TCID_50_ and incubated for 1 h at 4 °C. After adsorption, the monolayers were washed with DMEM without serum to remove the viral particles that were not adsorbed, then treated with 1 mL of the peptide solutions at the MNTC and incubated for 1 h at 37 °C. Later, the monolayers were rewashed with DMEM without serum, added to the cells, incubated for another 48 h at 37 °C, and then frozen for cell lysis, followed by plaque assay (item 2.5). Lastly, assays were performed to determine if the peptides could interfere with viral intracellular mechanisms. Therefore, Vero cell monolayers were inoculated with 100 µL of HSV-1 suspension at 100 TCID_50_ and incubated for 1 h at 37 °C. After this period, the monolayers were washed with DMEM without FBS and incubated for 1 h at 37 °C. Then, cells were treated with solutions of the peptides at the MNTC and incubated for 18 h at 37 °C to complete the viral replication process. After this period, cells were rewashed with DMEM without FBS and incubated for 24 h at 37 °C. Lastly, the microplate was frozen for cell lysis, followed by plaque assay (item 2.5). The post-penetration assay used the antiviral ACV as a control of inhibition. Afterward, the results were expressed in percentage of inhibition (PI). All assays were performed in triplicate [26,27].

### 2.7. Circular Dichroism

Circular dichroism (CD) experiments were carried out using a Jasco 815 spectropolarimeter (JASCO International Co., Ltd., Tokyo, Japan) coupled to a Peltier Jasco PTC-423L system for temperature control. In the experiments carried out in the presence of sodium dodecyl sulfate (SDS) micelles, 1 mM of peptide was added to 500 mM SDS in water. pH variations were also assessed, and samples were prepared to a final peptide concentration of 1 mM and 500 mM of SDS in acetate buffer 2 mM at pH 4, Tris–HCl buffer 2 mM at pH 7, and glycine-NaOH buffer 2 mM at pH 10. Peptides were also analyzed in trifluoroethanol (TFE)/water and pure water. Spectra were collected and averaged over five scans in the 190–260 nm spectral range, with 1 nm path length quartz cells, at 25 °C. A 0.2 nm step resolution, 10 nm.min^−1^ speed, 1 s response time, and 1 nm bandwidth were used [28].

### 2.8. Nuclear Magnetic Resonance

Initially, 300 μL of a suspension containing [I^5^, R^8^] mastoparan 2 mM, H_2_O/D_2_O 9:1, and 100 mM SDS-*d*_25_ was prepared for nuclear magnetic resonance (NMR) analysis. The NMR data were collected at 25 °C using a Bruker Avance III 500 NMR spectrometer running at 11.75 T (^1^H resonance frequency 500.13 MHz) and equipped with a maximum Z gradient of 55 Gauss/cm. The 2D ^1^H–^1^H total correlation spectroscopy (TOCSY) and nuclear Overhauser effect spectroscopy (NOESY) experiments were recorded and used with appropriate presaturation. They used a time-proportional phase increment (States TPPI/States) for quadrature detection in F1. The mixing times were set to 200 ms for TOCSY and NOESY experiments. TOCSY and NOESY data were processed using Topspin [13], and spectra manual assignment using the CcpNMR Analysis v.2.4 [29]. Structure calculation was performed using simulated annealing with the software ARIA [30] and CNS [31]. Lastly, UCSF Chimera [32] and CcpNmr Analysis [33] were used to evaluate structure quality.

### 2.9. Preparation of Large Unilamellar Vesicles

Large unilamellar vesicles (LUVs, ~100 nm diameter) were obtained by the extrusion of multilamellar vesicles (MLVs), as previously described [34]. The LUVs studied were made of pure POPC, POPC: POPS 2:1, POPC: Chol 2:1, POPC:Chol:POPS 70:5:25, or POPC:Chol:POPS 70:20:10. In all measurements, 10 mM HEPES buffer pH 7.4 with 150 mM NaCl was used.

### 2.10. Fluorescence Anisotropy

The fluorescence probes DPH and TMA-DPH were dissolved in 10 mM HEPES buffer pH 7.4 with 150 mM NaCl,. Then, 3 mM of LUVs were incubated for 30 min with each probe at a final concentration of 10 µM. After the incubation period, different concentrations of [I^5^, R^8^] mastoparan (5, 10, 15, 20, 25, 35, and 50 µg.mL^−1^) were added to the labeled LUVs and incubated for 40 min. The steady-state fluorescence anisotropy, ⟨r⟩, was calculated as follows, in Equation (2):(2)$$<r>\;=\frac{I_{VV}-{\mathrm{GI}}_{VH}}{I_{VV}+2{\mathrm{GI}}_{VH}}$$
where I*_VV_* and I*_VH_* are the parallel and perpendicular polarized fluorescence intensities measured with the vertically polarized excitation light and G = I*_HV_*/I*_HH_* is an instrumental correction factor accounting for the polarization bias in the detection system. Excitation/emission wavelengths were 350/432 nm for DPH and 355/430 nm for TMA-DPH. Fluorescence measurements were carried out using a Varian Cary Eclipse fluorescence spectrophotometer (Mulgrave, Victoria, Australia). The excitation and emission bandwidths were set to 5 nm and 10 nm, respectively, and the fluorescence spectra were recorded in quartz cuvettes with a path length of 0.5 cm at 37 °C. The measurements were repeated at least three times with independent measurements, ensuring the reliability of the results [35].

### 2.11. Statistical Analysis

The data’s mean and standard deviation (SD) were calculated from a minimum of three experiments. Non-linear regression was carried out for the dose/response curve, and two-way ANOVA for the mechanism of action assays. All statistical analyses were conducted using GraphPad Prism v. 8 and deemed significant when *p* < 0.05.

## 3. Results and Discussion

The discovery of peptide-based drugs has gained the interest of the scientific community. It is considered that this class of compounds may be used as a complementary therapy or, in some cases, as an alternative to small molecules [36]. A rich source of antiviral peptides is the venom of certain animals, such as arthropods. Initially isolated from wasp venom, the peptide mast-L has proven to be a promising template for designing new peptides [22,23,37,38]. The initial assays with the peptides were performed to determine cell viability and the MNTC (CC_20_—Cytotoxic Concentration for 20% of the cell monolayer or 80% of cell viability). The peptides mast-MO and mast-L showed over 90% cellular viability at 200 µg.mL^−1^. Therefore, the data did not enable us to determine the cytotoxic concentration for 50% of the cell monolayer (CC_50_ > 200 µg.mL^−1^). However, [I^5^, R^8^] mastoparan allowed 80% viability at 50 µg.mL^−1^ (Figure 1A). While the in vitro cytotoxicity data presented in Vero cells is encouraging, it is crucial to recognize the potential toxic effects of [I^5^, R^8^] mastoparan. Curiously, in a previous test with other cell lineages, including HEK-293, [I5, R8] mastoparan showed no cytotoxicity even at higher concentrations (LC_50_ > 200 µM). However, in a THP-1-derived macrophages’ culture, this peptide showed a much lower LC_50_ (24.5~12.9 µM) [22]. In recent assays using RAW 264.7 cells, the highest concentration (128 μM) of [I^5^, R^8^] mastoparan decreased cell viability. However, concentrations equal to or below 4 μM allowed for 96% cell viability [39]. This variation in cytotoxicity among cell lines emphasizes the need for further in vivo studies to comprehensively assess the safety profile of [I^5^, R^8^] mastoparan. Thus, the concentration of 50 µg.mL^−1^ was determined to standardize and compare the activities of the molecules in subsequent trials.

In vitro antiviral assays, mast-L, mast-MO, and [I^5^, R^8^] mastoparan at 50 µg.mL^−1^ induced a 72%, 86%, and 86% inhibition of HSV-1 replication (Figure 1B), respectively. The clinical standard antiviral acyclovir (20 µg.mL^−1^) was used as a positive control of viral inhibition. Afterward, serial dilutions of the peptides were tested against the virus. Following the dose/response curve, [I^5^, R^8^] mastoparan showed viral inhibition in a dose-dependent manner. At the lowest concentrations of 6.25 and 3.12 µg.mL^−1^, the PIs were 51% (SD = 14%) and 21% (SD = 15%), respectively. However, at a concentration of 12.5 µg.mL^−1^, the PI reached up to 75% (SD = 7%). at the highest concentrations, 25 and 50 µg.mL^−1^, [I^5^, R^8^] mastoparan inhibited viral replication up to 96% (SD = 3%) and 99% (SD = 1%), respectively. (Figure 1C). Furthermore, the defined effective concentration that inhibited 50% of viral replication (EC*_50_*) and the peptide’s selectivity index (SI) were approximately 6 and 8, respectively. The CC_50_ value used for the SI calculation was 50 µg.mL^−1^. At a concentration of 50 µg.mL^−1^, mast-MO exhibited the greatest inhibition. However, at 25 µg.mL^−1^ and below, the percentage of inhibition decreased to less than 20%, eventuallyreaching, reached 0%. Hence, the defined EC_50_ and SI of mast-MO were approximately 6.68 µg.mL^−1^ and >29.9 µg.mL^−1^, respectively (Table 1).

In the assays regarding the virus replication cycle, the results from mast-MO and [I^5^, R^8^] mastoparan suggested an inhibition during the initial stages of infection. More precisely, mast-MO showed 99% inhibition in the virus pre-treatment assay and entry step, whereas [I^5^, R^8^] mastoparan reached 99% viral inhibition in the virus pre-treatment assay in the adsorption and entry steps (Figure 1D). Indeed, virucidal activity is a common trait among antiviral peptides. These results suggest that [I^5^, R^8^] mastoparan and mast-MO can be classified as entry inhibitors, for which characteristics such as cationicity and hydrophobicity play a significant role, primarily against enveloped viruses [40,41]. Nevertheless, further assays with other enveloped and non-enveloped viruses are needed to address this matter.

Mastoparan 7 (MP7-NH_2_) showed activity against enveloped viruses: it presented broad-spectrum antiviral activity against enveloped viruses such as varicella-zosterVZV, vaccinia virus, cowpox virus, yellow fever virus, human respiratory syncytial virus, and HSV-1 [42]. Moreover, transmission electron microscopy analysis showed that VZV treated with MP7-NH_2_ had envelope damage, with its capsid separated from the envelope. Recently, a peptide named Smp76, purified from the scorpion *Scorpio maurus palmatus* venom, exhibited antiviral activity against hepatitis C and dengue viruses [43]. The authors stated that this peptide could prevent viral infection in cell cultures by neutralizing hepatitis C and dengue virions. In another study, a screening of peptides from the scorpion *Euscorpius validus* venom led to of identifying the antiviral peptide Eva1418, which is active against HSV-1 [44]. The authors successfully enhanced this peptide’s cell uptake and intracellular distribution by introducing histidine residues, which increased its α-helix-forming capability and amphiphilicity. In addition, spider venom-derived peptides have shown antiviral activity against different enveloped viruses, such as dengue, Zika [19,45], measles, and influenza viruses [46].

Some medically relevant viral diseases are caused by enveloped viruses (those with a membrane limiting the viral particle), such as influenza virus [47], HIV [48], flaviviruses (dengue [49], yellow fever, and Zika [50] viruses), SARS-CoV-2, MERS-CoV [51], and Ebola virus [52]. Viral envelopes are complex structures taken from the host cell membranes, with important viral glycoproteins incorporated into them [53]. Therefore, among other mechanisms of action, entry inhibitor peptides may exert their activity by viral membrane disruption [54,55], driving aggregation and interference upon envelope fusion [56]. AVPs may also interact with the surface viral protein during critical conformational changes [57,58,59]. To achieve this, the peptide’s secondary structure is an important feature. Numerous reports state mastoparans depend on an α-helix structure to exert their antimicrobial activity [60,61].

Since the molecule structures of mastoparan L (ID: 6DUL) and mastopan MO (ID: 6DUU) are already available and were previously well described [23], both CD and NMR analyses were carried out to elucidate [I^5^, R^8^] mastoparan structure in a membrane mimetic model. According to the CD analyses, [I^5^, R^8^] mastoparan assumes an α-helix conformation in the presence of micelles formed by 50% TFE (101.90%) and SDS-d_25_. In contrast, the peptide forms a random coil in water (34.27%) and in non-buffer dodecyl-phosphocholine (DPC) (51.79%) micelles. The observed positive band values close to 190 nm and minima near 208 and 222 nm are expected from α-helical structures. The highest α-helical content recorded was in the TFE/water mixture (1:1), indicating a stable secondary structure. The same behavior was observed when peptides were added to other hydrophobic/hydrophilic interfaces at different pH values, such as SDS-acetate (99.15%), SDS-Tris (90.60%), SDS-glycine (82.59%), and non-buffered SDS (94.49%) (Appendix A). These data show that [I^5^, R^8^] mastoparan maintains a stable α-helical structure even with pH variations.

The α-helical structure of [I^5^, R^8^] mastoparan was confirmed by structure calculation using NMR experiments in deuterated SDS-*d*_25_ micelles. Therefore, the NMR structure was determined from assembling 159 distance restrictions. TOCSY and NOESY analyses were performed simultaneously, and the peptide’s secondary structure was predicted by the chemical-shift index (CSI) (Figure 2B) using CcpNmr Analysis software v. 2.4 [33].

The peptide’s sequential assignment and spin system dispersion were evidenced in the TOCSY-NOESY HN-Hα superposition (Figure 2A). The superposition of a few signals imposed difficulty on the assignment process, such as 4K and 11K HN-Hα. However, the HN-HN correlation signals allowed us to identify the residues in positions 2–14. Despite this relatively low dispersion, the NOESY experiment presented substantial Hα-HN (i, i + 3) correlations, the first evidence of peptide folding (Figure 2B). CSI showed that [I^5^, R^8^] mastoparan assumed an α-helix structure between residues Asn-2 and Lys-12 (Figure 2B).

The 10 structures of lowest energy are shown in Figure 2C. The peptide was amphipathic in the presence of SDS micelles. The peptide presented four positively charged residues located on the same face and on the opposite face, which were the non-polar residues (Leu-3, Leu-6, Leu-7, Ala-10, and Leu-14) and seemed essential in the interaction with the membrane. [I^5^, R^8^] mastoparan has three well-defined helical turns with a highly hydrophobic N-terminus, whose residues seem to interact, which could be a peptide mechanism of peptide interaction with the membrane. The surface representation evidences the hydrophobic predominance of the peptide over the hydrophilic surface (Figure 2C). The biophysical characteristics of [I^5^, R^8^] mastoparan may be considered possible antiviral activity indicators.

[I^5^, R^8^] Mastoparan is cationic, with its +4 charge due to the presence of lysine and arginine residues in its primary structure, which can assume an α-helical conformation depending on the environment. Upon contact with the membrane lipid bilayer, mastoparans adopt an amphipathic α-helical structure with distinct hydrophobic and hydrophilic faces, inducing membrane destabilization (Figure 2C). However, when mastoparans encounter hydrophobic/hydrophilic interfaces, such as the one between bacterial membranes and the surrounding environment, they typically form α-helices [62].

Peptides from the mastoparan family and their analogs can interact with membranes, potentially disrupting bacterial membranes and viral envelopes [42,60]. However, evaluating such activity in a host cell/viral envelope scenario can be challenging, as the lipid composition of host cells and viral membranes is often quite similar. Therefore, selectivity may rely on differences in membrane properties such as charge, stiffness, or fluidity [63]. Based on the structural data of [I^5^, R^8^] mastoparan, it is inferred that this peptide may also affect lipid vesicles mimicking the viral envelope. Therefore, lipid vesicles (LUVs) labeled with fluorescent probes were set to interact with increasing concentrations of [I^5^, R^8^] mastoparan. These vesicles contained different compositions of lipids, mimicking biological membranes. According to the readings obtained from the LUVs labeled with the TMA-DPH and DPH probes, it was possible to observe that even at higher peptide concentrations, there were no changes in membrane fluidity, regardless of the lipid composition (Figure 3). These findings suggest that [I^5^, R^8^] mastoparan’s virucidal activity may not rely solely on disrupting membrane fluidity, and other properties like permeability or curvature could be involved. As indicated in a study with the peptide AH, such a mechanism has been described before, derived from the N-terminal domains of the hepatitis C virus non-structural protein 5A (NS5A) [63]. AH showed the ability to disrupt lipid vesicles the size of viral envelopes by sensing the high curvature of nanoscale vesicles. Additionally, the AH forms a tetrameric structure, which leads to pore formation [64]. Moreover, the lack of change across different lipid compositions may suggest a broad targeting mechanism, potentially affecting both host and viral membranes. Research on lipopeptides has indicated their potential antiviral effects by inhibiting the fusion of viral and cell membranes [65]. In a recent study, 11 cyclic peptides (CLPs) were tested on fusion membranes of varying compositions, stimulated by calcium, polyethylene glycol, and a fragment of the SARS-CoV-2 fusion peptide. The findings suggest that membrane composition plays a role in the CLPs’ ability to disrupt fusion. They also tested the antiviral activity of CLPs against SARS-CoV-2, and four of them (aculeacin A, anidulafungin, iturim A, and mycosubtilin) showed significant antiviral and antifusogenic properties [65]. Therefore, future assays are needed to address whether the peptide can and selectively target viral (but not host) membranes.

## 4. Conclusions

Developing new antiviral therapies is an elaborate task, as viruses have unique replication mechanisms and often mutate rapidly, making them difficult to target. Moreover, many viruses are poorly understood, contributing to the lack of effective treatments. Therefore, research efforts are imperative to identify new targets and develop novel antiviral strategies. In summary, we have shown the antiviral activity of three synthetic peptides derived from mastoparan. The mast-MO molecules and [I5, R8] mastoparan have shown potential in controlling HSV-1 infection. They may have an inhibitory effect on other viruses. The peptides derived from mastoparan are multifunctional and, hence, are valuable resources in the ongoing battle against viral infections. Further research into their mechanisms of action and interactions with viral envelopes holds considerable promise for developing effective antiviral treatments.

## Figures and Tables

**Figure 1 viruses-16-00948-f001:**
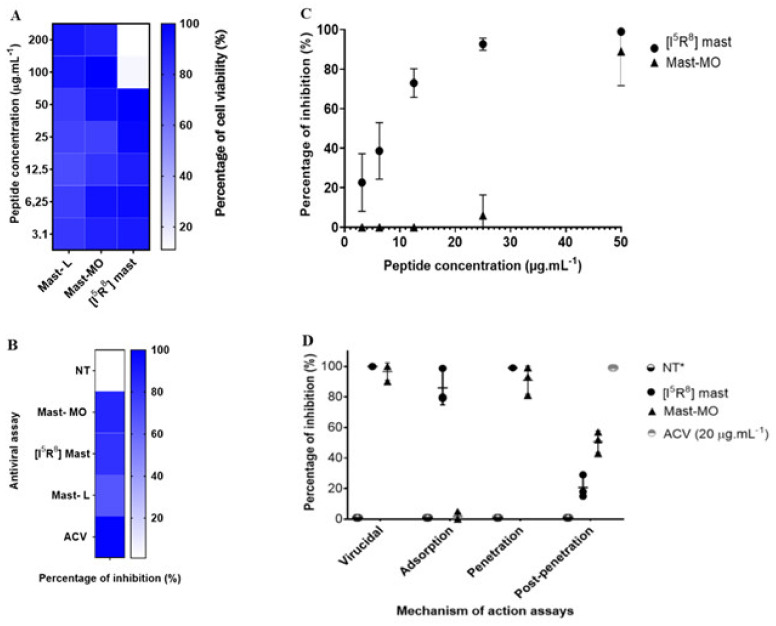
In vitro assays of the antiviral activity of the peptides mastoparan-MO, mastoparan-L, and [I^5^, R^8^] mastoparan. (**A**) Heat map showing MTT cell culture viability assays with the three mastoparans. (**B**) Antiviral triage assay with the three peptides. (**C**) Dose–response curves of the peptides mastoparan MO and [I^5^, R^8^] mastoparan. Acyclovir (ACV) was used as a positive control for inhibition (**D**) Time-addition assay of the peptides mastoparan MO and [I^5^, R^8^] mastoparan. The positive-negative control with only the cells infected with the HSV-1 was marked as not treated. * NT—Not treated.

**Figure 2 viruses-16-00948-f002:**
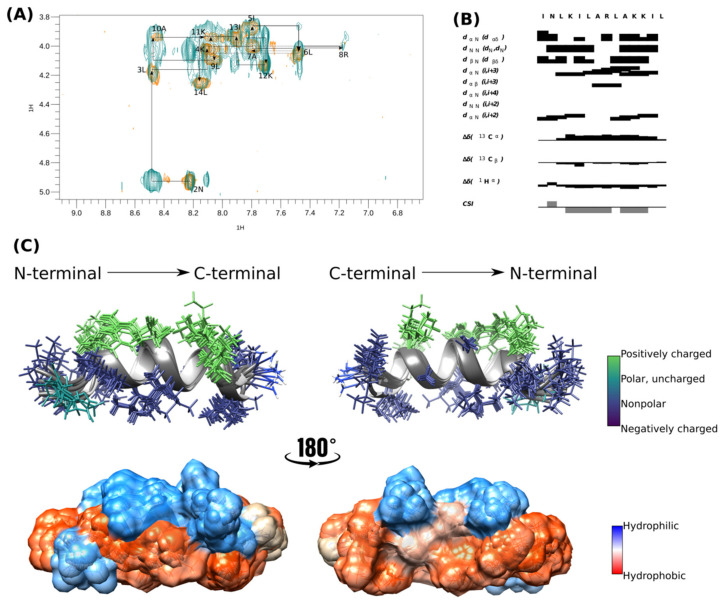
Tridimensional structure of [I^5^, R^8^] mastoparan obtained by NMR. (**A**,**B**) Analysis of the connectivity pattern for [I^5^, R^8^] mastoparan. (**C**) Overlapping of the ten structures with less energy and cloud model structure.

**Figure 3 viruses-16-00948-f003:**
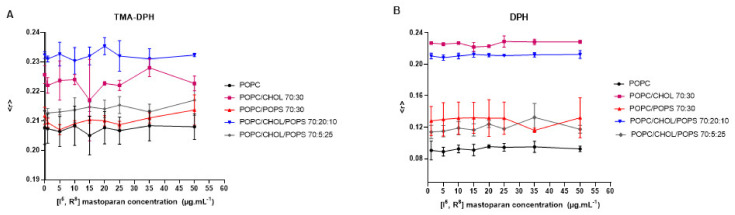
Interaction of the peptide [I^5^, R^8^] mastoparan with lipid vesicles labeled with TMA-DPH (**A**) and DPH (**B**), followed by fluorescence anisotropy measurements.

**Table 1 viruses-16-00948-t001:** Mastoparan peptides summary table.

Peptide	Initial Concentration (µg.mL^−1^) *	CC_50_ (µg.mL^−1^) **	EC_50_ (µg.mL^−1^) ***	SI ****
Mastoparan-L	200	>200	-	-
Mastoparan-MO	200	>200	6.68	>29
[I^5^, R^8^] mastoparan	200	>50	6.22	>8

* Concentration is defined based on the amount of each peptide available for the cytotoxicity assays. ** Cytotoxicity concentration, which presented 50% of cellular viability. *** Effective concentration that inhibited 50% of viral replication. **** Selectivity index.

## Data Availability

The data presented in this study are available on request from the corresponding author.

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
