# Peer review of "Antiviral Activities of Mastoparan-L-Derived Peptides against Human Alphaherpesvirus 1"

_viruses, 2024, doi:10.3390/v16060948_

Round 1

Reviewer 1 Report

Comments and Suggestions for Authors

Vilas Boas and colleagues present an analysis of the antiviral properties of mastoparan-L derived peptides against HSV-1 virions as well as their effects on virus attachment, penetration and replication. A structural analysis of [I5, R8] mastoparan was also performed. The manuscript is well written and the authors conclusions are supported by the data presented. This manuscript is well-aligned with the Viruses special issue on Antiviral Peptides. I have a few minor concerns.

1) Lines 148-160. The authors repeatedly state that the volumes of HSV-1 suspensions they are working with are 100 mL. This seems incorrect to me. Should 100mL be 100µL?

2) Line 298. “Most medically relevant viral diseases are caused by enveloped viruses….”. Rotaviruses, enteroviruses, rhinoviruses, papillomaviruses, noroviruses are MAJOR human viral pathogens that lack envelopes. The authors might consider rephrasing this inaccurate statement.

Reviewer 2 Report

Comments and Suggestions for Authors

 In this work “Antiviral activity of mastoparan-L derived peptides against human alphaherpesvirus 1” three peptides derived from Vespula lewisii venom were tested for their antiviral activity against herpes simplex virus type 1 (HSV-1) and possible mechanisms of action were investigated. However, the experiments and results described in the work are not sufficient to support the authors' conclusions. Therefore, I strongly recommend that this manuscript be reconsidered for publication only after a major revision.

Major Issues

1) The authors state that the characteristics of cationicity and hydrophobicity of mast-Mo and [I5, R8] mastoparan mean that they can be classified as entry inhibitors, primarily against enveloped viruses (row 279-282). However, in orther to say that, the antiviral activity of peptides should be tested on more enveloped viruses and at list one not enveloped viruses as control. The best thing would be to use another member of the herpesvirus family (for example HCM or HSV-2) and an enveloped virus belonging to another family (for example VSV). Only with these tests can they support their claim.

2) The authors used a temporal addition assay of the mast-Mo and [I5, R8] mastoparan peptides to identify which stage of viral replication they inhibit. This strategy is useful but not resolution. In order to discriminate between virucidal activity, adsorption inhibition or penetration inhibition specific assay should be performed (example of protocols doi: 10.3791/53124).

3) The Authors hypnotized that the virucidal activity of peptides belonging to the mastoparans family was due to an interaction with the membranes which potentially destroys the viral envelope. However, in the experimental condition presented in the manuscript, even at higher peptide concentrations the peptides were unable to change membrane fluidity, regardless of lipid composition. Author should discuss about this findings and tested alternative mechanism of interaction whit viral envelop, for example a protein–protein docking to test the interaction with HSV-1 glycoproteins.

 Minor Issues

1) Abstract/introduction: HSV-1 is the causative agent of cold sores, it can cause genital herpes but is mostly spreads by oral contact and causes infections in or around the mouth (https://www.who.int/news-room/fact-sheets/detail/herpes-simplex-virus).

2) Materials and Methods 2.6: To evaluate virucidal activity, the virus-peptide mixture should be incubated for 1h ant 37 °C and then diluted to the “subtherapeutic” (ineffective) concentration of the peptide under investigation before cell treatment (doi: 10.3791/53124). Moreover, the result should be compared with the inhibitory activity of the mixture virus-peptide diluted immediately (without incubation period) to a subtherapeutic concentration before infection. The authors should specify if this steps were done.

3) Fig 1A: I suppose that the values are “percentage of cell viability”. That should be specified in the figure.

4) Fig 1D: Which protocol-treatment were used for ACV treatment?

5) Fig 1D: To better explain the different treatment times used in the time addition test I suggest creating an image that schematizes the treatment protocols as a function of time.

6) I suggest to add a table with the IC50, EC50 and SI index values.

Reviewer 3 Report

Comments and Suggestions for Authors

Dear Authors,

After reviewing the manuscript, I have identified specific areas that require corrections and explanations.

Abstract, lines 18-20: Although HSV-1 can be sexually transmitted, this way of transmission is not the only one, and not the most common one either.  

Introduction, lines 38-40: HSV-1 is a virus, not a sexually transmitted infection. Genital herpes, most commonly caused by HSV-2, but sometimes also by HSV-1, is a sexually transmitted infection.

Introduction, lines 45 – 48: this sentence needs to be modified. There are clinically tested and approved treatment regimens for viral infections, including reactivations, associated with HSV-1 and HSV-2, which lead to viral clearance and elimination of symptoms. It is also true that it is impossible to eliminate the latent virus. However, having a latent HSV does not equal having a disease caused by a particular herpesvirus. Hence, the statement that “there is no record of a cure for the disease” is not accurate.

What was the TCID50 of HSV-1 used in this research?

Please include the origin of ACV in the methodology.

Results and discussion: lines 242 – 244:  “Thus, the concentration of 50 µg.mL-1 was determined to standardize and compare the activities of the molecules in subsequent trials.” I would advise presenting the CC50 as  >50 µg.mL-1 for [I5, R8] mastoparan and >200 µg.mL-1 for mast-MO and mast-L and then calculating the SI as above a certain value (e.g. SI > 7).

Lines 270 – 271: “The CC50 value used for the calculation was 50 µg.mL-1”, please explain the meaning of this sentence. To which calculations? Is it for the SI?

Lines 273 – 274: “Hence, the defined EC50 and SI of mast-MO were approximately 6.68 µg.mL-1.” There is no SI presented in this sentence.

I wish you could include an assay for the viral load, e.g. qPCR. I understand that it may not be possible to present the results of viral load measurement in this work. However, please consider including this in future experiments.

Comments on the Quality of English Language

The text would benefit from some minor language and style revisions to improve its clarity and coherence.

Reviewer 4 Report

Comments and Suggestions for Authors

The paper "Antiviral activities of mastoparan-L derived peptides against 2 human alphaherpesvirus 1" by Vilas Boas et al has been reviewed.

The authors describe the identification of mastoparan-derived antiviral peptides, with virucidal effect against HSV-1. The novelty of the paper is limited, because the anti-HSV-1 activity of different mastoparan peptides has been already reported in different papers.  Nevertheless, the paper is of interest, the research work is well described and the literature is adequate. The major issue is represented by the considerations about activity and toxicity. In fact, the selectivity index is below 10 and the IC90 is reached at a concentration close to the CC50 value. In my opinion, before further development, the authors should assess the absence of toxicity in vivo, using preliminary toxicity models such as insects. The discussion should be implemented considering the limitation of mastoparan derivatives and the potential toxic effects.  

Round 2

Reviewer 2 Report

Comments and Suggestions for Authors

The authors responded point by point to all comments. Although notable improvements have been made, the work remains at a very preliminary stage as the experiments suggested to implement it have not been performed. 

As stated by the authors in the coverletter, the present study mainly focused on the antiviral activity of mastoparan mast-MO and [I5R8] against HSV-1. Limited to this topic, the paper is complete and available for publication. However, I strongly suggest to the author to take the review comments into consideration when designing future work.

Reviewer 3 Report

Comments and Suggestions for Authors

Dear Authors,

Despite my previous comments, certain issues were not appropriately corrected.

Introduction, lines 37-39: You have corrected the sentence, but once again, I must underline that both HSV-1 and HSV-2 are viruses, not sexually transmitted infections. Please correct this; you could write that they may be responsible for sexually transmitted infections.

Introduction, lines 45 – 48: this sentence still needs to be modified. Once again, I must give the same comment as previously. There are clinically tested and approved treatment regimens for viral infections, including reactivations, associated with HSV-1 and HSV-2, which lead to viral clearance and elimination of symptoms. However, it is also true that it is impossible to eliminate the latent virus, but having a latent HSV does not equal having a disease. Hence, the statement that “there is no record of a cure for the disease” is not accurate. Also, citing an encyclopedia from 2009 to support such claims is highly unprofessional.

You have still not provided the TCID50 of HSV-1 used in this research! You have used a 100-fold TCID50 infectious dose in your experiments, which is correct, but what was the calculated TCID50 of virus stock? Please correct!

Thank you for adding the information on acyclovir. However, please add information on what was used to solubilize ACV – PBS, EtOH, water or DMSO. What was the concentration of the stock solution? If organic solvents were used, include information if their cytotoxicity and influence on antiviral experiments were tested.

Comments on the Quality of English Language

The text would benefit from some minor language and style revisions to improve its clarity and coherence.
